# Learning HTN Methods with Preference from HTN Planning Instances

**Zhanhao Xiao**[a]**, Hai Wan** [*a]**, Hankui Hankz Zhuo**[a]**, Andreas Herzig**[b]**, Laurent Perrussel**[c] **and Peilin Chen**[a]

[a]School of Data and Computer Science, Sun Yat-sen University, Guangzhou, China
[b]IRIT, CNRS, Toulouse, France
[c]University of Toulouse, Toulouse, France

## Abstract

The hierarchical task network (HTN) planning technique is used in a growing number of real-world applications. However in many domains, such as the logistics domain, as there exist thousands of cases, it is difficult and time-consuming for humans to specify all HTN methods to cover all desirable plans. This suggests that it is important to learn HTN methods to accomplish the tasks via decomposition. The traditional HTN-method learning approaches require complete executable plans and annotated tasks, which are often difficult to acquire in real-world applications. In this paper, we propose a novel framework to learn HTN methods from HTN instances with incomplete method sets and without annotated tasks. Besides, previous approaches demand total orders on the subtasks in the methods while our approach is capable of learning methods with partial orders. To reduce the number of methods learned, we consider priorities on methods and compute the minimal set of methods based on prioritized preferences. By taking experiments on three well-known planning domains, we demonstrate that our approach is effective, especially on solving new HTN problems.

## Introduction

The hierarchical task network (HTN) planning technique (Erol *et al.* 1994) is increasingly used in a number of real-world applications (Lin *et al.* 2008; Behnke *et al.* 2019). In the real-world logistics domain, such as Amazon and DHL Global Logistics, the shipment of packages is arranged via decomposition into a more detailed shipment arrangement in a top-down way according to the predefined decomposition methods. In practice, there exist a vast number of cases occurring, such as the delay caused by the weather, leading that it is difficult and time-consuming for humans to find all complete methods for all actions. This suggests that it is important to learn methods to help humans to improve the HTN domain.

Normally the domain experts have partially hierarchical domain knowledge, which possibly is not sufficient to cover all desirable solutions (Kambhampati *et al.* 1998). Different from classical planning which pursues an executable plan to achieve the declarative goal, the solution to the HTN planning problem requires to consider the hierarchical procedural goals, which are given by HTN methods. With partially hierarchical domain knowledge, a solution cannot be found via decomposition according to the given methods. One main reason lies in that the given method set is incomplete, which includes at least an incomplete method lacking subtasks. Keeping the hierarchical procedural knowledge, Geier and Bercher (2011) proposed a hybrid planning formalization, *HTN planning with task insertion (TIHTN planning)*, to allow generating plans via decomposing tasks according to the methods but also inserting tasks from outside the given methods. The following example shows an incomplete method set.

**Example 1.** *Consider an example in the logistics domain, suppose every task has only one method and a decomposition tree is shown in Figure 1. The initial task* ship(pkg1, whA, shopB) *is to ship a package from city A to city B and it has a method: to ship the package from the warehouse to the airport by truck, from city A to city B by plane and from the airport to the shop by truck. But in case the plane is not in the airport of city A, then the air transportation task cannot be accomplished. When arranging the plane to airport A,* fly(plane1, airpA)*, is done before loading to the plane, it generates an executable plan. If it is not allowed to insert actions, there is no plan to achieve the task* ship.

Actually, the plan with the inserted tasks offers a reference to accomplish the compound tasks, we refine the method by adding the inserted tasks. For example, the method of airShip is refined by adding fly as its subtask. By refining methods, we obtain new methods which generate the missing tasks, resulting in a new decomposition tree.

In practice, the missing of subtasks happens more likely on the methods of some compound tasks than on the methods of some other compound tasks. For example, the task ship is decomposed into the inter-city shipment and the intra-city shipment, while decomposing the task airShip varies with the place where the plane stays. It leads to a priority on the methods: some methods have a high priority to be refined. An excess of methods learned will slow down problem-solving, so we hope to learn as few refined methods

---

[*]Corresponding author.

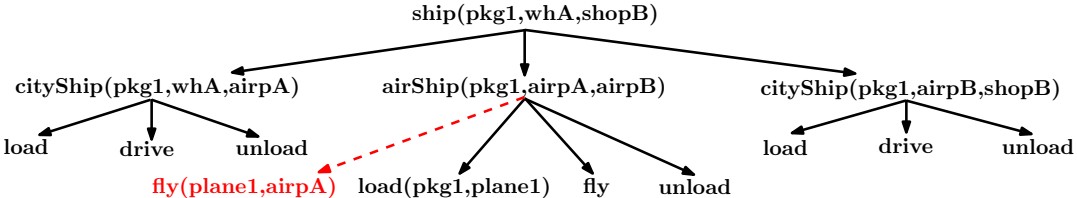

Figure 1: This is an Example of a decomposition tree from an incomplete HTN method set. The initial task ship(pkg1, whA, shopB) is decomposed into a sequence of primitive tasks (the black leaves) according to the original methods. But when the plane1 is not in airport A, the sequence is not executable. It becomes executable if arranging plane 1 to airport A before loading the package, which implies that fly(plane1, airpA) should be considered as a subtask of airShip.

as possible. The task is however challenging, as tasks can be inserted in various methods and an exponential number of method sets need to be considered.

The traditional approaches to learning HTN methods, such as (Hogg *et al.* 2008; Zhuo *et al.* 2014; Lotinac and Jonsson 2016), only concentrate on declarative goals and omit procedural knowledge obtained from the domain designer, which cannot be replaced simply by declarative goals. For example, every package needs a security check before being uploaded into the plane. If the action model is not complete, such as the 'check' action has not the effect 'checked', the declarative goal may not capture it. Besides, the approaches (Hogg *et al.* 2008; Zhuo *et al.* 2014) require the annotated preconditions and effects of tasks, which omit the hierarchical procedural goals and only consider the declarative goals like classical planning, so they require a complete executable plan as input. Whereas it is not a simple task to obtain complete plans, particularly when it involves thousands of situations. Furthermore, in many domains, it is difficult to verify the correctness of the annotations of tasks when they are taken as input. In this paper, we propose a novel framework to learn HTN methods from HTN instances with an incomplete method set, which always cannot generate executable plans only via decomposition. Besides, previous approaches restrict the tasks in the methods to be totally ordered, while we allow them to be partially ordered. Last but not least, we consider a prioritized preference on the methods learned.

Our contributions are listed as follows. First, we propose an approach to learning new methods by refining the original methods based on decomposition trees with task insertion. Second, we give a framework METHODLEARN to learn HTN methods from HTN instances with an incomplete method set. To reduce the number of methods learned, the method set learned by METHODLEARN is minimal w.r.t. a given prioritization. Third, we take experiments on three well-known domains and compare the percentage of solving new problems on our approach with method sets of different incompleteness and a classical learning approach, HTN-MAKER (Hogg *et al.* 2008). The experiment result shows that our approach is effective, especially on solving new HTN problems.

## Related Work

Besides those we mentioned above, there have been action model learning approaches related with our work. Garland *et*

*al.* (2001) proposed an approach to construct and maintain hierarchical task models from a set of annotated examples provided by domain experts. Similar to the annotated tasks, obtaining these annotated examples is difficult and needs a lot of human effort. Our work also is related to the works on learning the precondition of HTN methods (Ilghami *et al.* 2005; Xu and Muñoz-Avila 2005), which take the hierarchical relationships between tasks, the action models, and a complete description of the intermediate states as input. The similar work also includes (Nejati *et al.* 2006) and (Reddy and Tadepalli 1997), which used means-end analysis to learn structures and preconditions of the input plans. The precondition and effect of primitive actions can also be learned in (Zhuo *et al.* 2009). All these approaches to learning the precondition of methods require a complete method set as input.

The work on hybrid planning which combines classical planning and HTN planning is also related with our work. By relaxing the restriction of generating plans only via decomposition, Geier and Bercher (2011) proposed propositional TIHTN planning which allows to inserting primitive tasks to obtain executable plans. Later Alford *et al.* (2015) generalized it into lifted TIHTN planning by allowing variables in actions and predicates. In this paper, we focus on HTN planning and aim to learn HTN methods with the help of TIHTN planning.

## Problem Definition

We adapt the definitions of propositional HTN planning (Geier and Bercher 2011). For a propositional language $\mathcal{L}$, a state is a subset of the propositions in $\mathcal{L}$. In HTN planning, actions[1], noted $\mathcal{A}$, are classified into two categories: the actions the agent can execute directly are called *primitive actions* or *operators*, noted $\mathcal{O}$, while the rest are called *compound actions*, noted $\mathcal{C}$. Every primitive action $o$ is a tuple $(\mathsf{pre}(o), \mathsf{add}(o), \mathsf{del}(o))$ where $\mathsf{pre}(o)$ is a conjunction of literals called its precondition; $\mathsf{add}(o)$ and $\mathsf{del}(o)$ are sets of propositional symbols called its positive and negative effect. A primitive action $o$ is *applicable* in a state $s$ if $s \models \mathsf{pre}(o)$, which results in a state $\gamma(s, o) = (s \backslash \mathsf{del}(o)) \cup \mathsf{add}(o)$. A sequence of primitive actions $o_1, ..., o_n$ is *executable* in a state $s_0$ iff there is a state sequence $s_1, ..., s_n$ such that $\forall_{1 \leq i \leq n}, \gamma(s_{i-1}, o_i) = s_i$ and $o_i$ is applicable in $s_{i-1}$.

Given a set $R$, we use $\overline{R}$ to denote the set of all sequences over $R$ and use $|R|$ to denote the cardinality of $R$. For its

---
[1]"Action" is also called "task name".

subset $X$ and a function $f : R \longrightarrow S$, its restriction to $X$ is $f|_X = \{(r,s) \in f \mid r \in X\}$. For a binary relation $Q \subseteq R \times R$, we define its restriction to $X$ by $Q|_X = Q \cap (X \times X)$.

**Task networks.** A task network is a tuple $\mathsf{tn}=(T,\prec,\alpha)$ where $T$ is a set of tasks, $\prec \subseteq T \times T$ is a set of ordering constraints over T and $\alpha : T \longrightarrow \mathcal{A}$ labels every task with an action.

Every task is associated to an action and the ordering constraints restrict the execution order of tasks. A task $t$ is called *primitive* if $\alpha(t)$ is primitive, otherwise called *compound*. A task network is called *primitive* iff it contains only primitive tasks.

We say two task networks $\mathsf{tn} = (T,\prec,\alpha)$ and $\mathsf{tn}' = (T',\prec',\alpha')$ are *isomorphic*, denoted by $\mathsf{tn} \cong \mathsf{tn}'$, if and only if there exists a bijection $f : T \longrightarrow T'$ such that for all $t_1, t_2 \in T$, $t_1 \prec t_2$ iff $f(t_1) \prec' f(t_2)$ and $\alpha(t_1) = \alpha'(f(t_1))$, $\alpha(t_2) = \alpha'(f(t_2))$.

**HTN methods.** Compound actions cannot be directly executed and need to be decomposed into a task network according to HTN methods. Each *HTN method* $m=(c,\mathsf{tn}_m)$ consists of a compound action $c$ and a task network $\mathsf{tn}_m$ whose inner tasks are called *subtasks*. Note that a compound action $c$ may have more than one decomposition method.

In a task network, the decomposition is done by selecting a compound task, adding its subtask network and replacing it. The constraints about the decomposed task $t$ are propagated to its subtasks: the tasks before $t$ are before all its subtasks and the tasks after $t$ are after all its subtasks.

**HTN problems.** An HTN planning *domain* is a tuple $\mathfrak{D} = (\mathcal{L}, \mathcal{O}, \mathcal{C}, \mathcal{M})$ where $\mathcal{M}$ is a set of decomposition methods and $\mathcal{O} \cap \mathcal{C} = \emptyset$. We call a pair $(s_0, t_0)$ an *instance* where $s_0$ is the initial state and $t_0$ is the initial task. An HTN *problem* is a tuple $\mathcal{P} = (\mathfrak{D}, s_0, t_0)$.

In different literature, the solution to the HTN problem has different forms: mostly a plan (such as (Erol *et al.* 1994)), a primitive task network (such as (Behnke *et al.* 2017)) and a list of decomposition trees (such as (Zhuo *et al.* 2014)). In this paper, we consider a solution to the HTN problem as a decomposition tree rooted in the initial task $t_0$.

A decomposition tree is a tuple $\mathcal{T} = (T, E, \prec, \alpha, \beta)$ where $(T, E)$ is a tree, with nodes $T$ and with directed edges $E : T \longrightarrow \overline{T}$ mapping each node to an ordered list of its children; $\prec$ is a set of constraints over $T$; function $\alpha : T \longrightarrow \mathcal{A}$ links tasks and actions; function $\beta : T \longrightarrow \mathcal{M}$ labels every inner node with a decomposition method.

We use $\ll$ to denote the transitive closure of $\prec$ and the order defined by $E$. We say $t_1$ is a predecessor of $t_2$ if $t_1 \ll t_2$. Dually, we also say $t_2$ is a successor of $t_1$. According to $\ll$, we say the sequence constituted by the leaf nodes of $\mathcal{T}$ is its plan, denoted by $\vartheta(\mathcal{T})$.

**Definition 1** (**Valid decomposition trees**)**.** *A decomposition tree $\mathcal{T}$ is valid w.r.t. an HTN problem $\mathcal{P} = (\mathfrak{D}, s_0, t_0)$ iff its plan $\vartheta(\mathcal{T})$ is executable in $s_0$ and its root is $t_0$ and for every inner node $t$ where $\beta(t) = (c, \mathsf{tn}_m)$, it satisfies:*

*1. $\alpha(t) = c$;*

*2. $(E(t), \prec|_{E(t)}, \alpha|_{E(t)}) \cong \mathsf{tn}_m$;*

*3. if $(t, t') \in \prec$ then for every $st \in E(t)$, $(st, t') \in \prec$;*

*4. if $(t', t) \in \prec$ then for every $st \in E(t)$, $(t', st) \in \prec$;*

*5. there are no $t_1, t_2$ such that $t_1 \ll t_2$ and $t_2 \ll t_1$.*

**Solutions.** A solution to an HTN problem $\mathcal{P} = (\mathfrak{D}, s_0, t_0)$ is a valid decomposition tree $\mathcal{T}$ w.r.t. $\mathcal{P}$ and we say $(s_0, t_0)$ is solved under $\mathfrak{D}$ and is satisfied by $\mathcal{T}$.

**Example 2** (Example 1 cont.)**.** *If plane1 is already at airport A in $s_0$, the decomposition tree drawn with **black** arrows shown in Figure 1 is a solution to the HTN problem. $\sigma_1 = \langle load; drive; unload; load; fly; unload; load; drive; unload \rangle$ is its plan.*

**Method Learning.** In this paper, we assume that the original methods are kept as they come from the expert knowledge and they are sound in some situations. So, we only consider adding methods into the original domain. For an HTN domain $\mathfrak{D} = (\mathcal{L}, \mathcal{O}, \mathcal{C}, \mathcal{M})$ and a method set $\mathcal{M}'$, we use $\mathfrak{D}+\mathcal{M}' = (\mathcal{L}, \mathcal{O}, \mathcal{C}, \mathcal{M} \cup \mathcal{M}')$ to denote the resulting domain by adding $\mathcal{M}'$ into $\mathfrak{D}$.

An HTN method learning problem is defined as a tuple $(\mathfrak{D}, \mathcal{I})$ where $\mathfrak{D}$ is an HTN domain and $\mathcal{I}$ is a set of instances. A solution of the HTN method learning problem is a set of methods $\mathcal{M}'$ which should satisfy:

- all instances in the set $\mathcal{I}$ are solved under $\mathfrak{D}+\mathcal{M}'$;

- the learned method set $\mathcal{M}'$ is as minimal as possible;

- the learned methods in $\mathcal{M}'$ have as little inserted subtasks as possible.

## Refining Methods via Task Insertion

In this paper, we focus on the HTN problem with an incomplete method set, where there is no valid decomposition tree w.r.t. the problem. In other words, there is no executable plan obtained only by applying methods. By allowing inserting tasks, (Geier and Bercher 2011) proposes a hybrid planning formalization, TIHTN planning. A solution to the TIHTN problem is a TIHTN plan which is a primitive action sequence executable in the initial state and includes all primitive tasks obtained by applying methods and inserted primitive tasks. (Alford *et al.* 2015) gives a progression policy for TIHTN planning and it is not difficult to design a progression-based algorithm to find a TIHTN plan and a decomposition tree which excludes inserted primitive tasks.

Actually, the inserted tasks in the TIHTN plan are subtask candidates: they provide clues for refining the original methods by adding them as subtasks. Then, based on a TIHTN plan, we propose the completion profile to refine methods and complete decomposition trees.

Inspired by (Alford *et al.* 2015), we propose a progression-based algorithm to search TIHTN plans in Algorithm 1. First, we say a task is unconstrained in the current state if all its predecessors have been done and use uncons to denote the set of unconstrained tasks in the current state. In every step, we choose non-deterministiscally an unconstrained task to perform or decompose (line 4), where performance updates the state (line 9) and decomposition updates the tree (line 14-15). Once a task is performed or decomposed, it is labelled as 'done' (line 16). If the precondition of the primitive task chosen is not satisfied in the current

**Algorithm 1:** $\text{HPLAN}(\mathfrak{D}, s_0, t_0)$

**input** : An HTN domain $\mathfrak{D}$ and an instance $(s_0, t_0)$
**output**: A decomposition tree $\mathcal{T}$ and a plan $\sigma$

1  $s \leftarrow s_0$;     $\sigma \leftarrow \emptyset$;
2  $\text{uncons} \leftarrow T \leftarrow t_0$;     $E \leftarrow \emptyset$;
3  **while** $\text{uncons} \neq \emptyset$ **do**
4      choose non-deterministically some $t \in \text{uncons}$;
5      **if** $t$ *is primitive* **then**
6          **if** $s \not\models \text{pre}(\alpha(t))$ **then**
7              find a plan $\sigma'$ to $s'$ where $s' \models \text{pre}(\alpha(t))$;
8              $\sigma \leftarrow \sigma \circ \sigma'$;
9              $s \leftarrow s'$;
10         $\sigma \leftarrow \sigma \circ \alpha(t)$;
11         $s \leftarrow \gamma(s, \alpha(t))$;
12     **else**
13         choose non-deterministically
            $m = (c, \text{tn}_m) \in \mathcal{M}$ *s.t.* $\alpha(t) = c$;
14         $T \leftarrow T \cup T_m$ *s.t.* $\text{tn}_m = (T_m, \prec, \alpha_m)$;
15         $E \leftarrow E \cup \{t \times T_m\}$;
16     label $t$ is done;
17     update $\prec$ and uncons in $T$;
18     **if** *all* $t \in T$ *are done* **then**
19         **Return** $\sigma$ and $\mathcal{T}$

20 **Return** fail

---

state, it searches a plan to satisfy it (line 7) via an off-the-shelf planner, FF planner, which actually is a classical planning problem. When all tasks are labelled as done, it returns a TIHTN plan and a decomposition tree excluding inserted tasks.

**Example 3** (Example 2 cont.)**.** *If* plane1 *is not at airport A in* $s_0$*, the decomposition tree in Example 2 is not valid as its plan* $\sigma_1$ *is not executable in* $s_0$*. While* $\sigma_2 = \langle$load;drive;unload;fly;load;fly;unload;load;drive;unload$\rangle$ *is a TIHTN plan to the problem.*

### Refining Methods and Completing Decomposition Trees

Suppose the TIHTN planner outputs a plan $\sigma$ and a decomposition tree $\mathcal{T}$, we use $I_\sigma$ to denote all the inserted tasks in $\sigma$. The TIHTN plan actually is an ordering of primitive tasks and we extend the $\ll$ relation of $\mathcal{T}$ by considering the execution order of primitive actions in $\sigma$. To get the compound tasks, we use $N_\mathcal{T}$ to denote the inner nodes of the decomposition tree $\mathcal{T}$. Next, we show how to link these inserted tasks with the inner nodes $N_\mathcal{T}$ of the decomposition tree $\mathcal{T}$ to generate a new decomposition tree.

**Definition 2.** *We define a completion profile as a function* $\rho : I_\sigma \longrightarrow N_\mathcal{T}$*, such that for every inserted task* $t' \in I_\sigma$ *there is not a primitive task* $t_p \in \sigma$ *where either both* $t_p \ll \rho(t')$ *and* $t' \ll t_p$*, or* $\rho(t') \ll t_p$ *and* $t_p \ll t'$*.*

Intuitively, every inserted task is associated with a compound task as its subtask. Every inserted task is restricted to be performed before the predecessors and after the successors of its corresponding compound task.

Next, we define how to refine a method by inserting tasks. A completion profile leads to a set of refined methods by adding the relevant inserted tasks into the original methods. Formally, for a completion profile $\rho$, let $t$ be an inner node in the decomposition tree, we use $T_\rho^t = \{t' \mid \rho(t') = t\}$ to denote all inserted tasks associated with $t$. Then we use $T(\rho)$ to denote the range of function $\rho$, i.e., the inner nodes which have a non-empty set $T_\rho^t$. The inserted subtasks with the original subtasks of $t$ compose a new subtask network, written by $\text{tn}_\rho^t = (T_\rho^t, \ll|_{T_\rho^t}, \alpha_\sigma)$, where $\alpha_\sigma$ is the function $\alpha$ from the plan $\sigma$. Every non-empty set $T_\rho^t$ leads to a refined method $m_\rho^t = (c, (T_m \cup T_\rho^t, \prec_m \cup \ll|_{T_\rho^t}, \alpha_m \cup \alpha_\sigma))$ w.r.t. the original method $\beta(t) = m = (c, (T_m, \prec_m, \alpha_m))$. We use $\mathcal{M}_\rho$ to denote the set of refined methods from the completion profile $\rho$.

**Example 4** (Example 3 cont.)**.** *For the TIHTN plan* $\sigma_2$ *and the decomposition tree in Example 1, we have a completion profile* $\rho$ *where* $\rho(t_1) = $ airShip *and* $\alpha(t_1) = $ fly(plane1, airpA)*. The refined method is* (airShip, $(T'_m, \prec'_m, \alpha'_m)$) *where* $T'_m = \{$fly, load, fly, unload$\}$*.*

The completion profile actually completes the decomposition tree: the inserted tasks are connected with their corresponding inner nodes as their children. When we add new nodes into the decomposition tree, the integrity of ordering constraints will be destroyed. To avoid that, we define an operator closure to complete the ordering constraints. Formally, for a tree $\mathcal{T} = (T, E)$, we define its closure on the ordering constraint $\prec$ as $\text{closure}(T, E, \prec)$, which is given by:

$$\prec \cup \bigcup_{t \in T} \{(t', ch), (ch, t'') \mid ch \in E(t), t' \prec t, t \prec t''\}.$$

Intuitively, the closure operation completes the ordering constraints about the children which should be inherited from their parent.

Next we show how to complete the decomposition tree according to the completion profile.

We define the completion of the decomposition tree $\mathcal{T}$ by completion profile $\rho$ w.r.t. TIHTN plan $\sigma$ as $\mathcal{T}_\rho = (T', E', \prec', \alpha', \beta')$, which is given by:

$$T' := T \cup \bigcup_{t \in T(\rho)} T_\rho^t$$

$$E' := E \cup \{(t, st) \mid t \in T, st \in T(\text{tn}_\rho^t)\}$$

$$\prec' := \text{closure}(T', E', \prec) \cup \bigcup_{t \in T(\rho)} \ll|_{T_\rho^t}$$

$$\alpha' := \alpha \cup \alpha_\sigma$$

$$\beta' := (\beta \setminus \{(t, m) \mid t \in T(\rho)\}) \cup \{(t, m_\rho^t) \mid t \in T(\rho)\}$$

The procedure of completing a decomposition tree consists of first connecting the inserted tasks with the inner nodes, then completing the ordering constraints and finally updating the method applied as the refined method. The decomposition tree being completed will satisfy the instance:

**Proposition 1.** *Given an HTN problem* $\mathcal{P} = (\mathfrak{D}, s_0, t_0)$*, let* $\sigma$ *be one of its TIHTN plans and* $\mathcal{T}$ *be its corresponding decomposition tree and* $\rho$ *be one of their completion profiles. Then the completed decomposition tree* $\mathcal{T}_\rho$ *satisfies the instance* $(s_0, t_0)$ *under the new domain* $\mathfrak{D} + \mathcal{M}_\rho$*.*

*Proof.* First, we show that $\mathcal{T}_\rho$ is a valid decomposition tree w.r.t. $\mathfrak{D}+\mathcal{M}_\rho$. For every node $t$ in $\mathcal{T}_\rho$ with $\beta'(t) = (c, \mathsf{tn}_\rho)$, i) the function $\alpha$ is not reduced, so $\alpha'(t) = c$; ii) the edges between the task $t$ and its inserted tasks $T_\rho^t$ are added, so the task network induced by its children is isomorphic with $m_\rho^t$; iii) closure$(T', E', \prec)$ guarantees that all ordering constraints of $t$ are propagated to the inserted tasks and $\ll|_{T_\rho^t}$ only introduces the ordering constraints among the inserted subtasks in the same method, so conditions 3. and 4. are satisfied; iv) as the completion profile guarantees that no contradict pair about $\ll$ is introduced, condition 5. is satisfied.

Without removing nodes, the root of $\mathcal{T}_\rho$ is still $t_0$. As the plan $\vartheta(\mathcal{T}_\rho)$ is the TIHTN plan $\sigma$ executable in $s_0$, $\mathcal{T}_\rho$ satisfies the instance $(s_0, t_0)$. $\qquad\square$

When an HTN problem has incomplete methods, the completion profile offers a way to improve the HTN domain:

**Theorem 1.** *If an HTN problem $\mathcal{P}=(\mathfrak{D}, s_0, t_0)$ has a TIHTN plan but no solution, then there is a completion profile $\rho$ where the HTN problem $\mathcal{P}' = (\mathfrak{D}+\mathcal{M}_\rho, s_0, t_0)$ is solvable.*

*Proof.* Let $\sigma$ be a TIHTN plan of $\mathcal{P}$ with its decomposition tree $\mathcal{T}$. Suppose $\rho$ is a completion profile w.r.t. $\sigma$ and $\mathcal{T}$. By Proposition 1, the decomposition tree $\mathcal{T}_\rho$ satisfies the instance $(s_0, t_0)$ under the new domain $\mathfrak{D}+\mathcal{M}_\rho$. So, $\sigma$ is a solution of the HTN problem $\mathcal{P}'$. $\qquad\square$

When the completion profile only add decomposition methods, we have a corollary:

**Corollary 2.** *Every plan of the HTN problem $\mathcal{P} = (\mathfrak{D}, I)$ is also a plan of the HTN problem $\mathcal{P}' = (\mathfrak{D}+\mathcal{M}', I)$.*

*Proof.* As the original methods are still in the domain, the valid decomposition trees of the original problem $\mathcal{P}$ are also valid decomposition trees of the new HTN problem $\mathcal{P}'$. So, plans of $\mathcal{P}$ are also plans of $\mathcal{P}'$. $\qquad\square$

## Prioritized Preferences

To formalize the experience that the missing of subtasks happens more likely on some methods than other methods, we consider a priority on the methods. Generally, the priority comes from the confidences of domain experts on methods: the method believed to lack subtasks more likely to have a higher priority.

Given a method set $\mathcal{M}$, we define a prioritization as a partition on it: $P=\langle P_1, ..., P_n \rangle$ where $\bigcup_{1 \le j \le n} P_j = \mathcal{M}$. Intuitively, the decomposition methods in $P_i$ have a higher priority to be refined than those in $P_j$ if $i > j$. We further consider the prioritized preference in terms of cardinality.

Given a prioritization $P=\langle P_1, ..., P_n \rangle$ of $\mathcal{M}$, we consider the prioritized preference $\le_P$ as follows: for $\mathcal{M}_1, \mathcal{M}_2 \subseteq \mathcal{M}$, if there is some $1 \le i \le n$ such that

- $|\mathcal{M}_1 \cap P_i| \le |\mathcal{M}_2 \cap P_i|$ and

- for all $1 \le j < i, |\mathcal{M}_1 \cap P_j| = |\mathcal{M}_2 \cap P_j|$,

then we write $\mathcal{M}_1 \le_P \mathcal{M}_2$. We say $\mathcal{M}_1$ is strictly preferred over $\mathcal{M}_2$ w.r.t. $P$, written by $\mathcal{M}_1 <_P \mathcal{M}_2$, if $\mathcal{M}_1 \le_P \mathcal{M}_2$ and $\mathcal{M}_2 \not\le_P \mathcal{M}_1$.

## Preferred Completion Profiles

Generally, we hope to find a completion profile changing the original methods minimally under the prioritized preference.

We first define some notations: for a refined method $m_\rho^t$, we use $\tau(m_\rho^t)$ to denote its original method $m$. For a refined method set $\mathcal{M}'$, we use $\tau(\mathcal{M}')$ to denote all the original methods of the refined methods in $\mathcal{M}'$, i.e., $\tau(\mathcal{M}') = \{m \in \mathcal{M}| m = \tau(m'), m' \in \mathcal{M}'\}$. Note that several completions may be associated with the same decomposition method. For two decomposition methods $m_1'$ and $m_2'$, if $\tau(m_1') = \tau(m_2')$, we say $m_1'$ and $m_2'$ are *homologous*.

**Definition 3.** *Given a TIHTN plan and its decomposition tree, a completion profile $\rho$ is preferred w.r.t. preference $P$ if there is not a completion profile $\rho'$, such that $\tau(\mathcal{M}_{\rho'}) <_P \tau(\mathcal{M}_\rho)$.*

Intuitively, the preferred completion profile refines methods minimally under the prioritized preference.

Next, we will show how to find the preferred completion profile, as shown in Algorithm 2. First, we consider all inserted tasks in the plan as unlabelled (line 1). Then we scan all inner nodes from the nodes with a method of higher priority to the nodes with a method of lower priority (line 2-3). Next, for an inner node, we find the set of candidate subtasks $\Delta_t$ from the inserted tasks, which do not violate the ordering constraints if they were inserted as its subtasks (line 5). More specially, for the inner node $t$, the inserted tasks which are executed between the last task required to be executed ahead of $t$ and the first task required to be after $t$, are allowed to be added as subtasks of $t$. According to the total order $+$ in the decomposition tree, we define the subtasks candidate set $\Delta_t$ of $t$ as the set of the unlabelled inserted tasks between the last predecessor of $t$ and the first successor of $t$. Finally, we associate all tasks in the subtask candidate set to $t$ (line 5) and label them as subtasks (line 6). When all inserted tasks are labelled, it returns a preferred completion profile. It must terminate and the worst case is that the inserted tasks are associated with the root task.

Algorithm 2 only scan the nodes of the decomposition tree once and searching the subtask candidate set can be done in linear time, so the algorithm terminates in polynomial time.

---

**Algorithm 2:** COMPLETE$(\sigma, \mathcal{T}, P)$

**input** : A TIHTN plan $\sigma$, its decomposition tree $\mathcal{T}$ and a prioritization $P = (P_1, ..., P_n)$ on $\mathcal{M}$
**output:** A completion profile $\rho$

1   $I \leftarrow I_\sigma$;
2   **for** $j \leftarrow n$ *to* 1 **do**
3     **for** *each* $t \in N_\mathcal{T}$ *s.t.* $\beta(t) \in P_j$ **do**
4       **if** $I \neq \emptyset$ **then**
5         for every $t' \in \Delta_t \cap I$, set $\rho(t') = t$;
6         $I \leftarrow I \setminus \Delta_t$;

7   **return** $\rho$

---

Actually, to find a preferred completion profile, we only need to scan the inner nodes in the decomposition tree ac-

cording to the preference and link appropriate inserted tasks with inner nodes, which can be done in polynomial time.

Observe that the more detailed tasks are more sensitive to these situations and more easily to be thoughtless. There exists a class of HTN domains where actions can be stratified according to the decomposition hierarchy (Erol *et al.* 1996; Alford *et al.* ). In this case, we assume that an action is more abstract than its subtasks and we consider a preference in terms of a stratum-based priority: the more abstract actions have a lower priority to be refined.

## Learning Methods from Instances

As stated above, we only consider to introduce new methods into the HTN domain. However, an excess of methods introduced may slow down problem-solving significantly, as there are excessive choices to decompose tasks. To reduce the number of methods learned, we consider the minimal set of methods learned under the prioritized preference. We propagate the prioritized preference to the refined methods: if $\tau(m') \in P_j$ then $m' \in P_j$.

**Definition 4.** *Given a method set $\mathcal{M}'$ and its prioritization $P$, a subset $\mathcal{M}'_0$ of $\mathcal{M}'$ is the minimal set w.r.t. $P$ if there is not a subset $\mathcal{M}'_1$ of $\mathcal{M}'$ such that $\mathcal{M}'_1 <_P \mathcal{M}'_0$.*

To learn as few methods as possible, we first compute a preferred completion profile w.r.t. the stratum-based prioritized preference and then compute the minimal method set.

Suppose $\mathcal{M}'$ is a set of methods learned, we use $\mathcal{I}(\mathcal{M}')$ to denote the solvable subset of $\mathcal{I}$ w.r.t. $\mathfrak{D}+\mathcal{M}'$. Furthermore, we use $\overline{\mathcal{T}}(\mathcal{M}')$ to denote the set of decomposition trees w.r.t. $\mathfrak{D}+\mathcal{M}'$. The decomposition trees and the instances solved are monotonic w.r.t. the methods learned:

**Proposition 2.** *If $\mathcal{M}_1 \subseteq \mathcal{M}_2$, then $\overline{\mathcal{T}}(\mathcal{M}_1) \subseteq \overline{\mathcal{T}}(\mathcal{M}_2)$ and $\mathcal{I}(\mathcal{M}_1) \subseteq \mathcal{I}(\mathcal{M}_2)$.*

*Proof.* When $\mathcal{M}_1 \subseteq \mathcal{M}_2$, it means that there are more methods to be chosen to decompose compound tasks, in consequence there will be more decomposition trees generated.

As every HTN plan comes from the decomposition tree, if an instance $i \in \mathcal{I}(\mathcal{M}_1)$ has an HTN plan, then it has a decomposition tree $\mathcal{T}^i$ satisfying it, entailing that $\mathcal{T}^i \in \overline{\mathcal{T}}(\mathcal{M}_2)$. Thus, the instance $i$ also in $\mathcal{I}(\mathcal{M}_2)$. $\square$

**Method substitution.** The completion profiles from various instances may induce many refined methods which decompose the same compound action and generate similar executable plans. The vast increase in the number of methods will slow down the problem-solving significantly and we need to reduce the redundant refined methods which can be replaced by other methods. To compute the minimal set, we need to remove the redundant refined methods and define a method substitution operator.

**Definition 5.** *Given a decomposition tree $\mathcal{T}=(T, E, \prec, \alpha, \beta)$, let $T_{m_1}$ be the inner nodes with method $m_1$ and $m_2 = (c, (T_2, \prec_2, \alpha_2))$ be a homologous method with $m_1$. We define the decomposition tree that substitutes the method $m_1$ in $\mathcal{T}$ with $m_2$ as $\mathsf{sub}(\mathcal{T}, t, m') = (T', E', \prec', \alpha', \beta')$,*

*given by:*

$$T' := (T \setminus \bigcup_{t \in T_{m_1}} T_t^{\mathrm{add}}(m_1)) \cup \bigcup_{t \in T_{m_1}} T_t^{\mathrm{add}}(m_2)$$
$$E' := E|_{T'} \cup \bigcup_{t \in T_{m_1}} (\{t\} \times T_t^{\mathrm{add}}(m_2))$$
$$\prec' := \mathsf{closure}(T', E', \prec \cup \prec_2)$$
$$\alpha' := \alpha'|_{T'} \cup \alpha_2$$
$$\beta' := (\beta \setminus \{(t, m_1)|t \in T_{m_1}\}) \cup \{(t, m_2)|t \in T_{m_1}\}$$

*where $T_t^{\mathrm{add}}(m)$ denotes the inserted subtasks w.r.t. $t$ for the refined method $m$.*

After substituting a method $m_1$ with another homologous method $m_2$, if the resulting decomposition tree still satisfies the instance, it means that for this instance, the replaced method $m_1$ is redundant and can be replaced by $m_2$.

**Proposition 3.** *For two homologous methods $m_1, m_2$, let $\mathcal{T}' = \mathsf{sub}(\mathcal{T}, m_1, m_2)$. If $\mathcal{T}$ satisfies an instance $(s_0, t_0)$ and $\vartheta(\mathcal{T}')$ is executable in $s_0$, then $\mathcal{T}'$ satisfies $(s_0, t_0)$.*

*Proof Sektch.* It is not difficult to prove $\mathcal{T}'$ is a valid decomposition tree, which entails that it satisfies the instance. $\square$

Next, we generalize the notion of substitution into the decomposition tree set: given a decomposition tree set $\overline{\mathcal{T}}$ and two method sets $\mathcal{M}'_1, \mathcal{M}'_2$, we define the set of the decomposition trees that substitutes every occurrence of every method $m'_1$ in $\mathcal{M}'_1$ with some method $m'_2$ in $\mathcal{M}'_2$ which is homologous with $m'_1$, written by $\mathsf{sub}(\overline{\mathcal{T}}, \mathcal{M}'_1, \mathcal{M}'_2)$. With the substitution operator, we can reduce the refined methods:

**Proposition 4.** *Given an HTN domain $\mathfrak{D}$ and an instance set $\mathcal{I}$, let $\overline{\mathcal{T}}$ be its decomposition tree set, each tree of which satisfies its corresponding instance. For a refined method set $\mathcal{M}'$ and its subset $\mathcal{M}'_j$, if the plan of every decomposition tree in $\mathsf{sub}(\overline{\mathcal{T}}, \mathcal{M}', \mathcal{M}'_j)$ is executable in the corresponding initial state, then $\mathcal{I}(\mathcal{M}') = \mathcal{I}((\mathcal{M}' \setminus \mathcal{H}(\mathcal{M}'_j)) \cup \mathcal{M}'_j)$ where $\mathcal{H}(\mathcal{M}'_j)$ is the set of methods homologous with the methods in $\mathcal{M}'_j$.*

*Proof.* By Proposition 3, for every instance $i = (s_0^i, t_I^i)$ in $\mathcal{I}$ and its decomposition tree $\mathcal{T}^i \in \overline{\mathcal{T}}$, if $\vartheta(\mathsf{sub}(\mathcal{T}^i, \mathcal{M}', \mathcal{M}'_j))$ is executable in $s_0^i$, then it satisfies $i$. When the methods in $(\mathcal{H}(\mathcal{M}'_j) \setminus \mathcal{M}'_j)$ are substituted, every instance is satisfied w.r.t. the remaining methods. $\square$

**Theorem 3.** *Given a method set $\mathcal{M}'$ and its prioritization $P$, there exists a minimal subset $\mathcal{M}'_0$ of $\mathcal{M}'$ w.r.t. $P$ such that $\mathcal{I}(\mathcal{M}'_0) = \mathcal{I}(\mathcal{M}')$.*

*Proof.* As $\mathcal{M}'$ is finite, the minimal subset exists and in the worst case, $\mathcal{M}'$ itself is minimal. $\square$

Next we give an algorithm for the HTN method learning problem, as shown in Algorithm 3. The framework consists of two main components: the first iteration for learning methods by refining methods (line 2-8) and the second iteration for reducing refined methods (line 9-11). The first iteration first finds a TIHTN plan and its decomposition tree for every instance (line 3) by HPLAN and then computes

**Algorithm 3:** METHODLEARN($\mathfrak{D}, \mathcal{I}, P$)

**input** : An HTN domain $\mathfrak{D}$, an instance set $\mathcal{I}$ and a prioritization $P = (P_1, ..., P_n)$ on $\mathcal{M}$
**output:** A new method set $\mathcal{M}'$

1  $\mathcal{M}' \leftarrow \emptyset;\quad \overline{\mathcal{T}} \leftarrow \emptyset;$
2  **for** *each $i$ in $\mathcal{I}$* **do**
3  $\quad$ compute a plan and decomposition tree $(\sigma^i, \mathcal{T}^i) = \text{HPLAN}(\mathfrak{D}, i);$
4  $\quad \rho^i = \text{COMPLETE}(\sigma^i, \mathcal{T}^i, P);$
5  $\quad$ complete the decomposition tree $\mathcal{T}^i$ to $\mathcal{T}^i_\rho$ by $\rho^i$;
6  $\quad \overline{\mathcal{T}} \leftarrow \overline{\mathcal{T}} \cup \mathcal{T}^i_\rho;$
7  $\quad$ construct a new method set $\mathcal{M}^i_\rho$ from $\rho^i$;
8  $\quad \mathcal{M}' \leftarrow \mathcal{M}' \cup \mathcal{M}^i_\rho;$
9  **for** $j \leftarrow 1$ *to $n$* **do**
10 $\quad$ compute the minimal subset $\mathcal{M}'_j$ of $P_j[\mathcal{M}']$ s.t. every tree in $\text{sub}(\overline{\mathcal{T}}, \mathcal{M}', \mathcal{M}'_j)$ satisfies their corresponding instance;
11 $\mathcal{M}' \leftarrow \bigcup_{1 \le j \le n} \mathcal{M}'_j;$
12 **return** $\mathcal{M}'$

---

each preferred completion profile (line 4) by COMPLETE. Next, these decomposition trees are completed and a set of refined methods are constructed according to the completion profiles (line 5-8).

To reduce the refined methods, if the constants in the inserted subtasks are identical with the arguments in the subtasks and compound task, we use the corresponding variables to replace the constants in the inserted subtasks (line 8).

In the second iteration, we use a greedy strategy to find the minimal set: the refined methods with lower priority are reduced first, which is the opposite against the procedure of searching the preferred completion profile. Here we use $P_j[\mathcal{M}']$ to denote the refined methods in $\mathcal{M}'$ whose original methods are in the priority $P_j$. By Proposition 4, the refine methods in $P_j[\mathcal{M}']$ can be replaced by $\mathcal{M}'_j$ for every instance and $\mathcal{M}'_j$ is minimal in the priority $P_j$.

To pursue the refined methods with as few subtasks as possible, we take a breadth-first strategy to find the inserted tasks when computing TIHTN plans (line 7 in Algorithm 1).

In fact, our approach outputs a method set which sometimes may be a second-best solution for criterion 2 and 3 of the solutions, while it satisfies criterion 1:

**Theorem 4.** *Suppose $\mathcal{M}'$ is the method set learned by* METHODLEARN$(\mathfrak{D}, \mathcal{I}, P)$*, if every instance in $\mathcal{I}$ has a TIHTN plan under the domain $\mathfrak{D}$, then it also has a solution under the domain $\mathfrak{D} + \mathcal{M}'$.*

*Proof.* As every instance has a TIHTN plan, by Proposition 1, there exists a set of decomposition trees $\overline{\mathcal{T}}$, each of which satisfies each instance w.r.t. the domain $\mathfrak{D} + \mathcal{M}''$ where $\mathcal{M}''$ is a method set obtained via completion profiles (line 2-8 in Algorithm 3). Then $\mathcal{M}' \subseteq \mathcal{M}''$. By Proposition 4, each decomposition tree in $\text{sub}(\overline{\mathcal{T}}, \mathcal{M}'', \mathcal{M}')$ satisfies its corresponding instance in $\mathcal{I}$. Thus, every instance is solvable under the new domain $\mathfrak{D} + \mathcal{M}'$. $\square$

## Experimental Analysis

We have implemented Algorithm 3 based on Python 3.0 and developed an HTN method learner METHODLEARN. In this section, we evaluate METHODLEARN in three well-known planning domains comparing with HTN-MAKER (Hogg *et al.* 2008) on the learning performances.

We consider three domains which are evaluated on in HTN-MAKER (Hogg *et al.* 2008), *i.e.,* Logistics, Satellite, and Blocks-World, to evaluate our approach. We first get the problem generators from International Planning Competition website[2] and randomly generate 100 instances for each domain and take 75 instances as the training set and 25 instances as the testing set. We run METHODLEARN and HTN-MAKER with 75 instances growingly as input and obtain different learned method sets from these two approaches. The planning instance in the testing set is considered as solved, if its goal is achieved by a plan computed under the learned HTN method set via an HTN planner. For HTN-MAKER we use the well-known HTN planner SHOP2 (Nau *et al.* 2003) and for our approach METHODLEARN we still use Algorithm 1 without task insertion. The time bound of the HTN planner is set to 3600 seconds. In order to check if an instance is solved, we add a verifying action whose precondition is the goal and whose effect is empty in the last subtask of the initial task. The learning performance is measured via the percentage of the number of the solved instances on that of the testing instances, which is called the percentage of instance solved.

To simulate the incomplete method set as the input of METHODLEARN, we take the HTN domain description in the website[3] of SHOP2, and remove different sets of subtasks. Furthermore, to evaluate the influence of the different incompleteness of the given method sets on the learning performance, we consider three removal cases: 1) remove one primitive task from each method (if exists), with meaning the high completeness, noted by ML-H; 2) remove two primitive tasks from each method (if exists), noted by ML-M, with meaning the middle completeness; 3) remove one more compound task in some method of ML-M, noted by ML-L, with meaning the low completeness.

Consider the method set shown in Figure 1, For ML-H, we remove the first drive and the first fly in the methods cityShip and airShip, respectively, while for ML-M, we remove all drive and fly in the methods. For ML-L, the first cityShip is additionally removed from the method of ship based on the ML-M setting.

Figure 2 shows the learning performances of our approach and HTN-MAKER in the three domains. Generally, with the training set growing, the percentage of the problems solved increases, which does not violate Proposition 2. For the Logistics and Satellite domains, in the settings of ML-H and ML-M, METHODLEARN learns the necessary methods to solve all testing problems from a few instances. It is because the structure of these two domains is relatively straightforward and the decomposition trees still can be constructed by the incomplete method sets. In the ML-L setting, the com-

---

[2]http://ipc02.icaps-conference.org/
[3]https://www.cs.umd.edu/projects/shop/

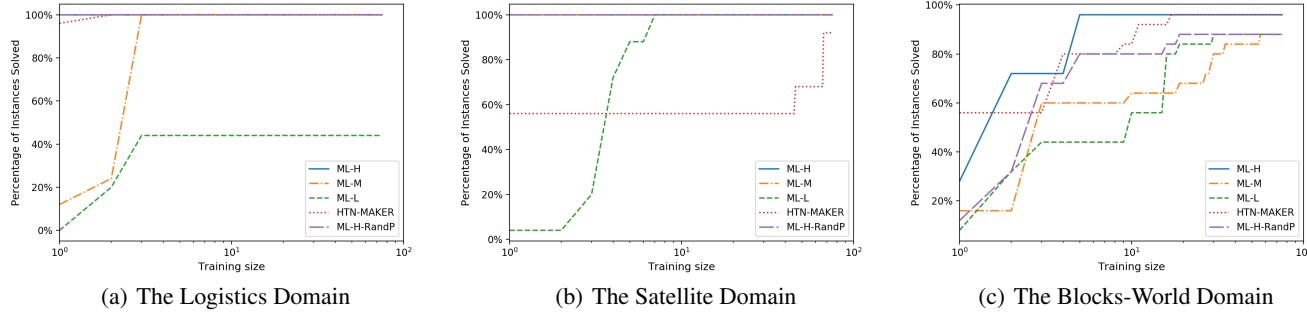

| (a) The Logistics Domain | (b) The Satellite Domain | (c) The Blocks-World Domain |

Figure 2: The Percentage of Solving Instances on Our Approach with Different Incomplete Method Sets and HTN-MAKER

pound action removed in the Logistics Domain, cityShip, contains more arguments, making the methods learned become more case-specific, which cannot contribute to other instances. For the Blocks-World domain, it cannot achieve the full convergence in each setting. The reason is that there are a few special instances which are significantly different from the training instances, resulting in that the methods learned hardly suit these special testing instances.

To evaluate our assumption on the stratum-based prioritized preference, we also compare it against a random prioritized preference, denoted by ML-H-RandP in Figure 2. When a completion profile associates the inserted tasks to a more abstract tasks, it generates a more case-specific method which may not suit other instances. It is shown that considering the stratum-based prioritized preference leads to a better learning performance.

## Discussion and Conclusion

We suppose that in the original method set, every compound action at least has a method to decompose. Our approach also can accept classical planning instances which only have a goal formula: we can trivially introduce a compound action of achieving the goal which is decomposed into a verification action whose precondition is the goal and whose effect is empty. Note that we only invoke a TIHTN planner to obtain plans for refining methods and focus on HTN problems. Also, the TIHTN planner needs to search actions to insert from the vast number of action candidates, a refined method including the missing subtasks helps to find the plan.

To sum up, we present a framework to learn HTN methods from HTN instances by refining methods. We also show that the methods learned by our framework are likely to solve new instances in the same classical planning domain. The experiment results demonstrate that our approach outperforms the traditional method learning approach, HTN-MAKER, given an appropriately incomplete method set as input. It is also illustrated that the stratum-based prioritized preference is effective.

## Acknowledgements

This paper was supported by the National Natural Science Foundation of China (No. 61573386), National Key R&D Program of China (No. 2018YFC0830600), Guangdong Province Natural Science Foundation (No. 2016A030313292), Guangdong Province Science and Technology Plan projects (No. 2016B030305007 and No. 2017B010110011), Guangzhou Science and Technology Project (No. 201804010435).

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
