# OpenReview forum: "Learning HTN Methods with Preference from HTN Planning Instances"
_icaps-conference.org/ICAPS/2019/Workshop/Hierarchical_Planning_

### Official Review · AnonReviewer1 · 2019-04-18
**Relevant paper. Significant contribution. Experiment section should be expanded.**

**Rating:** 8
**Confidence:** 3
**Overall Evaluation:** 2
**Significance:** 2
**Soundness:** 3
**Clarity:** 1

**Review:**

This paper is about learning HTN methods from planning instances. Its topic is very relevant to this workshop and the bibliography is to my best knowledge quite comprehensive. I did not completely check all the proofs but they seem ok to me. My main concern is about the experimental analysis and therefore the contribution significance. In my opinion, the testing protocol should be more strongly justified: why removing one primitive task from 2 methods? 2 primitive tasks from 2 methods etc. ? Why not 1 primitive tasks from 3 methods etc. ? How was HTN-MAKER evaluated? Moreover, it is not clear how the removed tasks and the instances in the training sets and in the testing sets were selected. Lastly, even though HTN benchmarks are not easy to generate, 3 domains are not enough to conclude. Did you experiment on other domains? Do you plan to extend the experimental analysis on other domains?

Minor typo: there existS a vast...
|= symbol in problem definition not clear to average reader
"I is a set of instance". Use/recall "I" notation also in "We call a pair..."
Clearify what "annotations" are in introduction section

**Reproducibility:**

3: authors describe the implementation and domains in sufficient detail

**Reviewer'S Confidence:**

3: medium

**Scholarship:**

3: excellent coverage of related work

---

### Official Review · AnonReviewer2 · 2019-04-20
**learning HTN methods**

**Rating:** 9
**Confidence:** 5
**Overall Evaluation:** 3
**Significance:** 3
**Soundness:** 3
**Clarity:** 2

**Review:**

The paper introduces a technique to learn new methods to extend an existing HTN planning problem. The new methods are learned based on a set planning problems, which allow an TIHTN solution. I.e., for each problem instance, one TIHTN solution is generated (i.e., a solution in which the insertion of tasks is allowed). Based on this solution, the *inserted* tasks are merged back into the decomposition tree that is part of the TIHTN solution thereby effectively extending existing methods. How the model is altered, i.e., into which methods the tasks get inserted, is decided based on a user-specified prioritization of the existing methods. These altered methods are added as additional methods to the original HTN planning problem thereby allowing to find HTN solutions for all given problem instances (even if the respective problems were not solvable before).

While the paper has several minor language flaws (please fix them for the final version), it is still very well written with regard to explaining its content (only in some parts more examples would be beneficial, but there are already many given). Content-wise the paper obviously advances the state of the art (given the authors didn't miss any significant related work in the literature), so it's clearly a strong accept.

Please find below a comprehensive list of corrections and suggestions. They are not ordered by priority. Corrections/etc. that I deem important are thus marked as such.

-
as noted above, there are many language issues. I did not make the effort marking every one of them, so please read the paper carefully or let if proof-read. The following list is surely incomplete:
* "another compound tasks" (task)
* "hope to learn as minimal completed". I believe this to be wrong. Maybe "as few"?
* "where D is an HTN domain, I is a set of instances." Use "and" instead of comma.
* "than one decomposition methods." (method)
* "and we extend << relation of T" (the << relation)
* "so the condition 3. and 4. are" ("so conditions 3. and 4. are")
* "to scan the inner node in the decomposition tree" (nodes)
* "The huge expansion of method set will" (unclear what this means; incorrect English)


-
some math formulae seem to be set incorrectly, e.g., "tn=" in the first line of "Task networks" seems to be standard text instead of being in math mode. Also, some spacing between relations (e.g., \prec) seems wrong many times (spaces are missing).

-
if I got it right, decomposition trees are witnesses to solvability, which means that they have to contain orderings that might not be present in the task hierarchy, but that got added during search. This is however not specifically stated, but probably should as this differs from its standard definition. Further, where are these additional orderings be put into? Are they put into \prec or into <<? (I think this should be stated as well).

-
Alg.1:
* in lines 4 and 11 you use the expression "choose". I recommend to use "choose non-deterministically" instead to make perfectly clear that this choice point is a backtrack point (i.e., the choice can be wrong; note that there are algorithms for which there are also non-non-deterministic (deterministic) choice points, i.e, where every choice is correct, so this makes clear that this is not the case here).
* line 7. I was wondering (a) which plan is found (any plan? or an optimal one? is this maybe an interesting unanswered question to find better "expanded" hierarchies?) and (b) *how* it is found (probably this does not matter, right? I guess you just use a classical planner there?)
* I found it slightly confusing that in several lines you used two expressions in one line. (And space should not be an issue here)
* line 11 is missing the criterion that alpha(t)=c.
* line 12 although it should be clear to everyone that T_m is the task set of tn_m, this is not explicitly stated/introduced. I an not sure whether this should be done.
* line 13 does not read like pseudo code, but like a comment. Only after reading line 15 it becomes clear that it's actually meant as pseudo code. I would probably formalize this somehow.
* line 16: I would add the actual tuple behind T, as otherwise one might wonder why T is returned although it was not even used in the code (T is never mentioned there, only it's elements). Then it will also become clear which elements have not yet been defined. beta, for instance, has to be constructed according to line 11 (which is not mentioned).

-
Well, despite my many suggestions for Alg.1 I am still wondering why you even care writing this down. The algorithm actually gives nothing new with regard to related work (if I hot it right). It's actually just standard TIHTN progression search as defined by Alford et al. with explicit mentioning how a decomposition tree is constructed. The latter is completely uninteresting, since every planner would (and is) constructing this anyway, so I see zero value for the community. In fact, I first had to spend minutes trying to read/understanding the code until I figured out that it's nothing new. This time could be saved... I would at least move it to an appendix I suppose.

-
"In every step, we choose an unconstrained task to perform
or decompose (line 4), where performance updates the state
(line 9) and decomposition updates the tree (line 12)."
I strongly believe that both the terms "perform a task" and even more "performance" are wrong. I vote towards changing it to "to apply..." and "application". If you change it, make sure to change it everywhere. It does not only appear in this sentence.

-
This comment is the one that is most important to me: I had real problems (and spent way too much time) understanding Def. 2 and would recommend to improve it in several ways.
* I believe it will be much clearer if you make clear that the mapping maps to the *inner* nodes of the tree. You could to this in text form (before or in the definition), but you can also define a new subset of T (e.g., T'\subset T being defined as the inner nodes of T) and define the mapping based T' rather than on T.
* Second, I recommend to explain for each of the tasks where exactly they come from. E.g., when you use t' the first time in Def.2, write t'\inI_\sigma. When you use t_p the first time, write "t_p in sigma.
* It was not clear to me where the restriction that you pose (i.e., the last two half lines in Def.2) come from and what they imply. I.e, (a) are they used to obtain a more "plausible" completion or (b) are they in fact *required* as otherwise something would not work anymore? This should *definitely* be explained, as it's an important information. Closely related: Are these criteria "soft" enough that a completion always exists or can one construct planning domains so that a completion cannot exist according to this definition? (Maybe this information can be added as well?)
* Further, the explanation below Def. 2 is still rather abstract. Can you add a graphic that illustrates the additional criteria that are described at the end of Def.2?
* I also think it would help to mention early on that you describe later how to find such a completion and for now only give the definitions that define their properties.

-
You define a completed method as the union of the original task network and a new one. This way, there cannot exist an ordering between the original task network and the newly inserted tasks. Is this really on purpose? This seems strange to me. I think later you add orderings to a decomposition tree, but this does not help, since the tree only follows from the model, but it does not influence it.

-
Example 4 looks strange. You give an example for a method, but the task network is a set with four elements, which is syntactically not correct. Is this some (non-introduced) short variant of a totally unordered task network?

-
In the definition of closure you use the string "st", which probably stands for "sub task", right? This is not introduced. Further, in the formal definition these tasks are called "children", so I would call them "ch" instead.

-
You invest quite some time and formalizations to construct an adapted decomposition tree (e.g. with the closure definition and the ones below). Why do you do this? After you constructed new methods you can simply say: "let T be a decomposition tree resulting from the application of the methods sequence ... (which now includes the new methods)". Or am I wrong here? This seems much easier and less technical. Then, Prop.1 would become obsolete. However, instead you would have to show that the new sequence of methods will lead to a solution. But maybe this is easier than proving the decomposition tree properties.

-
I would definitely delete Cor.2. Not only is it *absolutely trivial* (of course plans cannot suddenly become invalid just because the domain gets larger), but this has also nothing to do with any of the paper's content. Mentioning this triviality really seems strange to me.

-
Is N_T in line 3 in Alg.2 defined?

-
I would rename "Proof" of Prop.3 into "Proof sketch" as you only explain how one *would* prove it.

-
I have a major concern about the evaluation and would strongly recommend to address this in a final version:
From my point of view, it's not sufficiently explained what models HTN-Maker produces. Thus, it's also not clear what you evaluate. If the models are not equivalent (with regards to the plans they can possibly produce), what worth does such a comparison have?
To illustrate my concerns consider the following situation: Comparing the solving time of blocksworld-1 with logistics-200 is completely pointless, as they have nothing to do with each other and might be completely differently hard.
In your case, the differently learned problems probably have "something to do with each other", but what exactly is it? Why do we compare solving times of two differently learned models if we do not know that they are equivalent? And if they are equivalent, then I believe the evaluation only shows which kind of model is better for the used planner.
Thus, I ask to make clear:
(a) what your purpose of the evaluation is,
(b) how the models produced by HTN-Maker and your's differ and in which way they serve the same purpose, and
(c) how the evaluation has even any worth at all if the models are not equivalent (this is essentially the same as the question what properties the learned models share so that it makes sense to compare them even if the solution sets are not the same)

**Reproducibility:**

2: some details missing but still appears to be replicable with some effort

**Reviewer'S Confidence:**

4: high

**Scholarship:**

3: excellent coverage of related work

---

### Official Review · AnonReviewer3 · 2019-04-23

**Rating:** 9
**Confidence:** 5
**Overall Evaluation:** 3
**Significance:** 3
**Soundness:** 3
**Clarity:** 3

**Review:**

The paper presents a new mechanic to learn HTN methods from actual plan traces.
The authors aim not at learning the full model, but to complete a partial model -- which is a very reasonable assumption.
Learning is based on a partial decomposition tree that explains only a part of the plan trace. It is generated using a TIHTN planner.
Using this plan, the authors assign the tasks in the trace that are not explained by the decomposition tree to inner nodes in a consistent (Def 2) way. This is assignment is done via preferences, which in turn are based on the decompositional structure of the problem.
Lastly, the authors present an empirical evaluation of their work.

Overall
=======
The paper is overall well written, understandable and clearly presents its ideas. The technical part is a bit hard to follow, but this is expected and probably not avoidable. I especially like that completion profiles are clearly separated from their computation, which creates a nice framework for future research.
The idea is novel and very interesting. It paves the way for future investigations into this topic. Especially the idea of using a TIHTN planner for generating partial decomposition trees and using the decomposition hierarchy to compute preferences is very nice.
The only large issue I have with the paper is with the evaluation: in contrast to the rest of the paper, it is somewhat confusingly written. Some of its sentences seem to be ungrammatical. The figures contain probably too many different planners -- I would separate the evaluation of the reversed ordering of sample traces into a new figure. This one seems to be a bit full.



Questions
=========
- computing the preferences is based on the fact that the domain is tail recursive. Do you have any idea how to do this in the case it is not?



Suggestions
===========

- To understand Def 2, I had to draw a picture of the relative ordering of t, t' and \rho(t). It might be useful for the reader to include it into the paper.

Title: I think the "with Preference" might, just might, be misleading. My first impression when reading it was that the authors want to learn an HTN model that includes preferences - which they don't. I know that this reading is grammatically wrong, but it might shape the first opinion of a reader. It might be worthwhile to rephrase it so that it is immediately clear that you learn general HTN methods using preferences to determine where to add tasks.


Evaluation:
The evaluation is significantly harder to read than the rest of the paper and seems to to stem from the same authors. Several details are hinted at ("we randomly generate 100 instances") but the details are explained only later (that these instances stem from different pics of locations for trucks and packages). Some sentences sound really strange "To capture the original ...", "delivered with the truck transportation".

Concerning the TIHTN planner:
You use an algorithm inspired by Alford'15. I assume that the following two modifications might improve the speed of that planner and thus the overall performance of the learning technique. It might be worth experimenting with it.
1) Hoeller (ICAPS'18) recently proposed an updated version of SHOP's original HTN algorithm, which reduces the branching significantly (bottom line: only branch on one abstract task)
2) I don't think it is necessary to find a full plan in line 7. Instead you could consider a modification that adds just one action to the head of the plan. This way you essentially integrate the classical planner into the hierarchical one, which might be more efficient if you use a heuristic search version of HPLAN (see Hoeller'18). Also it should reduce the overhead for calling a classical planner in (I assume) almost all search nodes.
Currently, you use BFS (p7, left column) for the HPlan algorithm. Using Hoeller's algorithm you can use A* with an admissible heuristic, which should speed up the search.


Partial Parses: Bartak (ICAPS'18) proposed a technique for plan verification in HTN planning based on parting attribute grammars. Instead of using HPLAN, you could use their technique to compute a partial decomposition tree. In addition you might get partial parses for tasks not explained by the main decomposition tree. These partial parses end in some abstract task from the hierarchy. Instead of adding the unexplained primitive tasks to methods you could consider adding these abstract tasks -- which might improve your technique as you can now infer more complex models.



Minor Issues
=================
- there are several situations where you use citations as nouns, e.g. "(A and B 2009) showed that". It reads better if you use "A and B (2009) showed that"
- p1, left column: I would cite, Behnke et al AI Communications 2019 instead of Biundo et al. 2011.
- p1, left column: "decomposing into" -> "decomposition into"
- p4, left column: the union of two task networks is never formally defined.
- p5, left column: "sensitive to situations" -> "sensitive to these situations."
- p6, Proposition 4: "each [tree] satisfies each instance" -> shouldn't each tree satisfy its respective instance?
- p6, right column: The reference to algorithm 1 seems to be weird. It might either be Algorithm 3 or some Definition.
- p7, right column: "which confirms Proposition 2". Technically it only does not contradict (proof by example (in most cases) is impossible)
- p8, right columns: Hogg 2018, AAAI-=8 should read AAAI18.

**Reproducibility:**

3: authors describe the implementation and domains in sufficient detail

**Reviewer'S Confidence:**

4: high

**Scholarship:**

3: excellent coverage of related work